Association between different types of preoperative anemia and tumor characteristics, systemic inflammation, and survival in colorectal cancer

Zhou Chaoxi 1
Ma Hongqing 1
Wang Guanglin 1
Liu Youqiang 1
Li Baokun 1
Niu Jian 1
Zhao Yang 2
Wang Guiying wangguiying@hebmu.edu.cn 3
1 Second Department of General Surgery, Fourth Hospital of Hebei Medical University , Shijiazhuang , China
2 Department of Research, Fourth Hospital of Hebei Medical University , Hebei , China
3 Department of General Surgery, Second Hospital of Hebei Medical University , Shijiazhuang , China
Dong Peixin
Electronic publication date: 2023 Dec 21
Publication date: 2023
Volume: 11
Electronic Location ID: e16293
Received 2023 Aug 17; Accepted 2023 Sep 22
Copyright: ©2023 Zhou et al.
Copyright year: 2023
Copyright holder: Zhou et al.
License: This is an open access article distributed under the terms of the Creative Commons Attribution License, which permits unrestricted use, distribution, reproduction and adaptation in any medium and for any purpose provided that it is properly attributed. For attribution, the original author(s), title, publication source (PeerJ) and either DOI or URL of the article must be cited.
License URL: https://creativecommons.org/licenses/by/4.0/

Keywords: Anemia, Inflammation, Colorectal cancer, Risk factors, Prognosis

Funding: Health Commission of Hebei Province 20210042 This work was supported by the Health Commission of Hebei Province (No. 20210042). The funders had no role in study design, data collection and analysis, decision to publish, or preparation of the manuscript.

==============================
Background

Patients with colorectal cancer often have anemia and other symptoms after diagnosis, especially in patients with advanced colorectal cancer. This study explored the association between different types of preoperative anemia and tumor characteristics and inflammatory response in patients with colorectal cancer and to evaluate the prognosis of patients with different types of anemia before operation.

Methods

The clinical data of 95 patients with colorectal cancer treated in the Fourth Hospital of Hebei Medical University from February 2016 to January 2018 were retrospectively analyzed. According to the hemoglobin concentration (Hb), mean corpuscular volume (MCV), mean hemoglobin content (MCH) and mean hemoglobin concentration (MCHC), the patients were divided into the non-anemia group, normal cell anemia group, and small cell anemia group. The three groups’ general data, oncological characteristics, and mGPS scores were compared. The patients were followed up for five years, and the survival analysis was carried out. The cox proportional hazard regression model was used to analyze the prognostic factors of patients with colorectal cancer.

Results

The preoperative anemia rate of patients with colorectal cancer was 43.15% (41/95). There were significant differences in gender, weight loss, CA724, tumor location, tumor size, TNM stage, mGPS score, and positive expression rate of Ki-67 among different anemia groups. There was a significant difference in survival time among a non-anemia group, small cell anemia group, and normal cell anemia group (P < 0.05). Multivariate analysis showed that tumor size, TNM stage, distant metastasis, mGPS score, Ki-67 positive expression rate, and anemia type were independent risk factors affecting the prognosis of colorectal cancer patients (P < 0.05).

Conclusion

The oncological characteristics of colorectal cancer patients with different types of preoperative anemia are different. Preoperative anemia and systemic inflammatory status are independent risk factors for the prognosis of colorectal cancer patients.

Introduction

Colorectal cancer (CRC) is the world’s third most common malignant tumor and the second most common cause of cancer-related death (Han et al., 2021). Patients with colorectal cancer often have anemia and other symptoms after diagnosis, especially in patients with advanced colorectal cancer (Sinicrope, 2022). Anemia is a common disorder in CRC, with 40%–60% in colon cancer patients and 20%–51% in individuals with rectal cancer (Chapman et al., 2019). The forecast of CRC depends primarily on the tumor stage at diagnosis. With the increase of tumor focus area, the possibility of acute and chronic blood loss increases, and advanced patients are prone to malnutrition anemia when they suffer from tumor consumption for a long time (Gvirtzman et al., 2021). Numerous studies have suggested that the higher the pathological stage of CRC, the higher the probability of anemia (Lipton et al., 2021; Väyrynen et al., 2018). A recent study has highlighted that anemia incidence in CRC patients with Dukes stage C and D is up to 60%–75% (Deng, Weng & Zhang, 2023). According to statistics, the incidence of anemia in patients with gastrointestinal tumors in China is 51.13%, and more than 90% of them have not been given any treatment to correct anemia before finding a malignant tumor (Ciardiello et al., 2022). Thus, there is an urgent clinical need to explore the role of anemia in patients with CRC. However, few studies have evaluated the relationship between clinical and pathological characteristics of colorectal cancer patients and different types of anemia.

Emerging evidence indicates that some patients with colorectal cancer will have a systemic inflammatory consumption reaction, manifested by increased C-reactive protein (CRP) and decreased albumin, thus causing chronic inflammatory anemia (Hampel et al., 2022). Chronic inflammatory anemia is related to circulating cell inflammatory factors, which can be detected in infections, rheumatic diseases, and malignant tumors (Marques, Weiss & Muckenthaler, 2022). Many studies have reported the association between anemia and poor prognosis in colorectal cancer patients (Weng et al., 2022). However, more data are needed to confirm the relationship between essential parameters such as anemia, tumor characteristics, and systemic inflammation. Patients with CRC are usually accompanied by anemia and chronic inflammation symptoms. The most common is microcytic hypochromic anemia caused by iron deficiency (other uncommon causes include thalassemia and chronic disease anemia) (Phipps, Brookes & Al-Hassi, 2021). The causes of normal and large cell anemia are more diverse. Among them, microcytic hypochromic anemia caused by iron deficiency not only increases the demand for allogeneic blood transfusion during the perioperative period but also is closely related to the poor prognosis of tumor patients (Berg et al., 2023). Furthermore, normocytic anemia might increase the demand for perioperative blood transfusion and the incidence of postoperative complications (Holtedahl et al., 2021). Moreover, there is no report on the relationship between different types of anemia and prognosis in patients with colorectal cancer before operation. Therefore, we collected retrospectively collected the medical records of CRC patients in our hospital and carried out a retrospective analysis to explore the relationship between different types of preoperative anemia and characteristic tumor parameters and systemic inflammatory reaction in patients with CRC and explored the prognosis of CRC patients with different types of preoperative anemia.

Materials & Methods

General data

Throughout the electronic medical record, 95 patients who received surgical treatment in our hospital from February 2016 to January 2018 were chosen according to the inclusion criteria: (1) Patients who were diagnosed with CRC for the first time and underwent radical surgery; (2) pathologically confirmed as CRC; (3) patients who received the relevant index test within one week before the operation; (4) there are relatively complete clinical data and follow-up related information; exclusion criteria: (1) patients with or are suffering from other malignant tumors; (2) patients with blood system diseases; (3) patients with severe liver, kidney, heart, and brain dysfunction; (4) patients with preoperative radiotherapy, chemotherapy, or immunotherapy; (5) patients who have been treated with hormones, hemopoietic drugs or blood transfusion within three months; (6) incomplete clinical information and follow-up data. All samples obtained in this study were approved by the ethics committee of the Fourth Hospital of Hebei Medical University and abided by the ethical guidelines of the Declaration of Helsinki. The ethics committee agreed to waive informed consent (Approval number: 2018ME0156).

Blood sample collection

The blood samples of patients were collected within one week before the operation. Indicator determination: a Mindray BC-6800 automatic blood cell analyzer was used to determine the whole blood cell count. An automatic immunoluminescence analyzer detected the tumor markers.

Patient grouping

Blood routine examination is an essential method for the diagnosis of anemia. The hemoglobin concentration (Hb), mean corpuscular volume (MCV), mean hemoglobin content (MCH), and mean hemoglobin concentration(MCHC) were crucial items in the blood routine examination. According to the Hb, MCV, MCH, and MCHC of patients before the operation, the included patients of this retrospective study were divided into the non-anemia group, normal cell anemia group, and small cell anemia group. Ninety-five patients were included in the study, including 54 males and 41 females. The age range was 49∼78 years old.

Data collection

Preoperative data of qualified patients, including general information, hematological indicators, clinicopathological characteristics, and preoperative anemia status were obtained from the medical record. The patients’ hematological indicators, such as tumor marker indicators, serum albumin, and C-reactive protein (CRP) indicators were assessed for mGPS scoring. CRC staging was based on the 7th edition of the 2009 International Union against Cancer TNM staging of colorectal cancer (Amin et al., 2017). The positive rate of Ki-67 was evaluated according to the immunohistochemical staining results, and the critical value was 25%.

Primary outcome

The primary outcome in this study was the comparison of preoperative anemia status and tumor marker index of participants. The method outcome was the survival of colorectal cancer patients with different preoperative anemia types.

Follow-up methods

The main ways of follow-up were outpatient reexamination, telephone, and chat software. The follow-up period was calculated from the day of the operation and was followed up every three months after the operation. The follow-up contents were abdominal ultrasound, blood, biochemical, and tumor markers every three months after the operation. CT comparative examination was performed every six months (adjusted according to the specific situation), and enterostomy was performed every 6–12 months (adjusted according to the patient’s condition). The deadline for follow-up was December 2022, and the overall survival time was defined as the date from the operation to the last effective follow-up or the date of death of the patient.

Statistical analysis

SPSS 22.0 (SPSS Inc., Chicago, IL, USA) and PRISM.10 software (GraphPad Software, La Jolla, CA, USA) were used for data analysis. The continuous variables were expressed as the mean   ±  standard deviation (SD). The categorical variables were represented as percentages. The chi-square test, One-way ANOVA with Dunnett’s post-hoc was employed to assess the statistical significance among the three groups. Multivariate analysis was carried out by the Cox’s regression multiple hazard model. The survival curve was drawn using the Kaplan–Meier method, and the survival analysis was performed using the Log-rank test. The Cox proportional risk regression model was used to analyze the prognostic factors of colorectal cancer patients. Cox regression analysis has been widely used in the field of survival analysis. Cox proportional hazards regression is a semi-parametric survival analysis method which can be used to study the relationship between multiple covariables and the arrival time of an event. P < 0.05 was regarded as statistically significant.

Results

General information for participants

According to the results of a preoperative blood test, there were 52 cases in the non-anemia group, 14 in the small cell anemia group, and 29 in the normal cell anemia group. There were statistically significant differences in gender, weight loss, and BMI among the three groups (P < 0.05, Table 1).

Table 1 Comparison of the general information of participants in different groups.

Item	n	Non anemia (n = 52)	Microcytic anemia (n = 14)	Normocytic anemia (n = 29)	t/ χ2	P	
Age (yr, Mean ± SD)		56.42 ± 6.05	59.70 ± 6.33	59.86 ± 6.70	0.960	0.075	
Gender					6.874	0.021*	
	Male	54	37	4	13			
	Female	41	15	10	16			
Age(years)					2.125	0.212	
	<65	34	20	6	8			
	≥65	61	32	8	21			
Weight loss(kg)					7.962	0.014*	
	≤5	55	37	7	11			
	>5	40	15	7	18			
BMI (kg/m2 , Mean ± SD)			29.02 ± 3.21	27.45 ± 2.66	27.57 ± 2.80	1.204	0.003*	
								
Smoking		19	4	7	8	1.336	0.068	
Drinking		28	7	10	13	0.903	0.870	
Notes.

BMI Body mass index

P-values were obtained using χ2 tests for categorical variables. Wilcoxon test was used for continuous variables.

The asterisk implies statistical significance.

Comparison of clinicopathological characteristics of participants in different groups

Among patients with different types of preoperative anemia, there were significant differences in the tumor size, TNM stage, and Ki-67 positive expression rate among the three groups (P < 0.05). However, there was no significant difference in tumor location, vascular tumor thrombus, nerve invasion, tumor differentiation, distant metastasis, and lymph node metastasis ( P > 0.05, Table 2).

Table 2 Comparison of clinicopathological characteristics of participants in different groups.

Item	n	Non anemia (n = 52)	Microcytic anemia (n = 14)	Normocytic anemia (n = 29)	χ 2	P	
Tumor location					8.597	0.072	
	Colon	57	25	11	21			
	Rectum	38	27	3	8			
Tumor size(cm)					9.142	0.006*	
	<5	59	30	10	19			
	≥5	36	22	4	10			
Vascular tumor thrombus				1.873	0.296	
	No	62	27	12	23			
	Yes	33	25	2	6			
Nerve invasion					1.633	0.339	
	No	88	51	13	24			
	Yes	7	1	1	5			
Tumor Differentiation					1.885	0.345	
	Well-moderate	60	29	11	20			
	Poor-undifferentiated	35	23	3	9			
TNM stage					13.837	0.001*	
	I–II	51	37	4	10			
	III–IV	44	15	10	19			
Distant metastasis					2.896	0.198	
	No	74	47	8	19			
	Yes	21	5	6	10			
Lymph node metastasis				3.125	0.154	
	No	43	29	4	10			
	Yes	52	23	10	19			
Ki-67 expression rate				8.004	0.011*	
	≤25%	32	25	1	6			
	>25%	63	27	13	23			
Notes.

P-values were obtained using χ2 tests for categorical variables.

The asterisk implies statistical significance.

Comparison of preoperative anemia status and tumor marker index of participants

The preoperative anemia rate of colorectal cancer patients was 43.15% (41/95) in this study. Among patients with different types of anemia before the operation, there were statistically significant differences in mGPS scores among the three groups (P < 0.05, Table 3). However, there was no significant difference in CEA, CA199, CA242, CA724, and CA125 (P > 0.05, Table 3). The percentage of patients with CA199 (+) and CA199(−), different mGPS scores, and CEA(−) and CEA (+) among the three groups were demonstrated in Figs. 1, 2 and 3.

Table 3 Comparison of preoperative anemia status and tumor marker index of participants among the three groups.

Item	n	Non anemia (n = 52)	Microcytic anemia (n = 14)	Normocytic anemia (n = 29)	χ 2	P	
CEA (µg/L)					2.339	0.196	
	(−)	71	43	10	18			
	(+)	24	9	4	11			
CA199(U/mL)					1.435	0.326	
	(−)	83	47	13	23			
	(+)	12	5	1	6			
CA724(U/mL)					6.871	0.291	
	(−)	81	52	10	19			
	(+)	14		4	10			
CA242(U/mL)					0.765	0.572	
	(−)	79	46	11	22			
	(+)	16	6	3	7			
CA125(U/ mL)					1.284	0.423	
	(−)	76	47	10	19			
	(+)	19	5	4	10			
	mGPS score					0.970	0.000*	
	0	39	31	1	7			
	1	35	11	7	17			
	2	21	10	6	5			
Notes.

P-values were obtained using χ2 tests for categorical variables.

The asterisk implies statistical significance.

Figure 1 The percentage of patients with CA199(+) and CA199(−) among the three groups.

Figure 2 The percentage of patients with different mGPS scores among the three groups.

Figure 3 The percentage of patients with CEA (−) and CEA (+) among the three groups.

Survival of colorectal cancer patients with different preoperative anemia types

The follow-up time of 95 patients was 6–60 months, the median follow-up time was 43 months, and the average survival time was 37 months. The 5-year survival rate of the normocytic anemia group was better than the microcytic anemia group. After the Log-rank test, the difference in survival time among the two groups was statistically significant (P = 0.038, Fig. 4).

Figure 4 Survival of colorectal cancer patients with different preoperative anemia types.

Univariate analysis of prognosis of colorectal cancer patients undergoing radical surgery

The results of the univariate analysis showed that tumor marker CA199, tumor size, TNM stage, distant metastasis, lymph node metastasis, mGPS score, Ki-67 positive expression rate, anemia type, and other aspects were related factors affecting the prognosis of colorectal cancer patients undergoing radical surgery (P < 0.05, Table 4). However, gender, age, weight loss, tumor markers CEA, CA724, CA242, CA125, tumor location, tumor differentiation, presence or absence of vascular tumor thrombus, and presence or absence of nerve invasion were not related factors affecting the prognosis of colorectal cancer patients (P > 0.05, Table 4).

Table 4 Univariate analysis of prognosis of 95 patients undergoing radical surgery.

Item	Assignment	n	HR	95%CI	P	
Gender					0.134	
	Male	1	54	1.984	0.732∼1.312		
	Female	0	41				
Age(years)					0.065	
	<65	0	34	1.659	0.665∼2.031		
	≥65	1	61				
Weight loss(kg)					0.042*	
	≤5	0	55	1.582	0.787∼2.862		
	>5	1	40				
CEA (µg/L)					0.117	
	(−)	0	71	1.117	0.732∼3.585		
	(+)	1	24				
CA199(U/mL)					0.021*	
	(−)	0	83	1.209	1.018∼2.323		
	(+)	1	12				
CA724(U/mL)					0.072	
	(−)	0	81	1.437	0.865∼3.412		
	(+)	1	14				
CA242(U/mL)					0.295	
	(−)	0	79	0.865	0.584∼0.537		
	(+)	1	16				
CA125(U/ mL)					0.131	
	(−)	0	76	1.012	0.652∼3.124		
	(+)	1	19				
Tumor location					0.302	
	Colon	0	57	0.743	0.543∼3.050		
	Rectum	1	38				
Tumor size(cm)					0.011*	
	<5	0	59	2.003	0.524∼3.081		
	≥5	1	36				
Vascular tumor thrombus				0.112	
	No	0	62	2.211	0.685∼3.119		
	Yes	1	33				
Nerve invasion					0.214	
	No	0	88	1.396	0.665∼4.003		
	Yes	1	7				
Tumor differentiation					0.158	
	Well-moderate	0	60	1.957	0.875∼3.114		
	Poor-undifferentiated	1	35				
TNM stage					0.001*	
	I–II	0	51	3.324	2.359∼18.876		
	III–IV	1	44				
Distant metastasis					0.012*	
	No	0	74	4.565	6.565∼59.642		
	Yes	1	21				
Lymph node metastasis					0.038*	
	No	0	43	2.382	1.815∼2.998		
	Yes	1	52				
mGPS score					0.034*	
	0	0	39	2.784	6.881∼28.262		
	1	1	35				
	2	2	21				
Ki-67 expression rate					0.026*	
	≤25%	0	32	3.498	14.993∼55.473		
	>25%	1	63				
Types of anemia					0.005*	
	No anemia	0	51	3.034	8.896∼44.233		
	Microcytic anemia	1	13				
	Normocytic anemia	2	31				
Notes.

The asterisk implies statistical significance.

Multivariate analysis on the prognosis of colorectal cancer patients undergoing radical surgery

Multivariate analysis showed that tumor size, TNM stage, distant metastasis, mGPS score, Ki-67 positive expression rate, and anemia type were independent risk factors affecting the prognosis of colorectal cancer patients (P < 0.05, Table 5).

Table 5 Multivariate analysis of prognosis in 95 cancer patients.

	β	sx−	Wald	P	OR	95%CI	
CA199	0.732	0.382	1.352	0.201	0.985	0.786∼5.741	
Tumor size	1.462	0.197	3.654	0.044*	2.655	3.182∼7.225	
TNM stage	1667	0.692	6.188	0.010*	4.019	1.079∼22.848	
Lymph node metastasis	1.015	0.624	3.014	0.072	1.876	0.955∼9.967	
Distant metastasis	2.558	0.769	20.562	0.001*	6.782	2.565∼69.581	
mGPS score	1.532	0.564	4.055	0.029*	4.883	1.533∼13.846	
Ki-67 expression rate	1.601	0.199	10.551	0.001*	1.898	3.352∼7.542	
Types of anemia	1.325	0.382	8.536	0.002*	3.015	1.912∼9.367	
Notes.

The asterisk implies statistical significance.

Discussion

Patients with colorectal cancer usually have anemia and chronic inflammation (Weng et al., 2022). The most common cause is small cell hypochromic anemia caused by iron deficiency (other uncommon causes include thalassemia and chronic disease anemia). In contrast, the causes of normal and large cell anemia are more diverse (Patel, 2022). In this study, we observed an interaction between systemic inflammation and malignant tumors. There is a consensus that long-term chronic inflammation will cause anemia in the body (El Ghouayel et al., 2022). Inflammatory factors induce the production of excessive iron, which closes the iron output channel and reduces iron absorption in the gastrointestinal tract (Altintas et al., 2022). In this study, 95 patients with CRC were grouped according to different types of anemia. The relationship between oncological characteristics and systemic inflammatory reaction of patients with different preoperative anemia was evaluated, and the prognosis of patients was assessed as well.

Previous studies have shown that the decrease of hemoglobin in the blood of patients with colorectal cancer, especially normal cell anemia, is related to systemic inflammation (Jocić et al., 2022). Jocić et al. (2022) studied the pathogenesis of colorectal cancer and found that the development of intestinal dysfunction promoted bacterial translocation, leading to systemic inflammation (Schneider et al., 2018). This study also found that different types of anemia in colorectal cancer were related to the mGPS score. As we all know, the mGPS score is an inflammatory prognosis score, which can reflect chronic inflammation status by evaluating the body’s C-reactive protein and serum albumin (Hacker et al., 2022). This study shows a significant difference in the mGPS score between the non-anemia group and the two types of anemia group. Therefore, systemic inflammation is an essential determinant of hemoglobin in patients with colorectal cancer (Zhao et al., 2018). Giudice & Selleri (2022) showed that small cell anemia was related to tumor T stage and tumor location, so they believed that tumor size and TNM stage could determine a tumor load in the body, and excessive load would also cause the reduction of hemoglobin. Similarly, this study also found that different types of anemia were closely related to tumor location, tumor size, and TNM stage, consistent with Giudice & Selleri’s (2022) study. Auer et al. (2018) also found that compared with the distal colon and rectum, the hemoglobin of patients with proximal colon tumors was significantly lower. Disturbance of colonic microbiota can lead to the occurrence and development of colorectal cancer and interfere with the absorption of nutrients (Gomes et al., 2022).

Iron is the most important factor causing anemia, which is necessary for both prokaryotic and eukaryotic organisms to have cell functions (Chaparro & Suchdev, 2019). Most pathogenic bacteria have enhanced iron capture mechanisms, so they compete with protective bacteria for free iron (Jamwal, Sharma & Das, 2020). The intestinal flora of the proximal colon is abundant. More iron and other nutrients are absorbed than that of the distal colon, and the intestinal flora of men is more active than that of women (Deivita et al., 2021). Therefore, this is consistent with the conclusion of this study. Different types of anemia are closely related to gender and tumor location (Peake & Noguchi, 2022). Early research on the prognostic significance of blood Hb levels in patients with colorectal cancer has been controversial. It has been reported that the prognosis of colorectal cancer patients with anemia during the perioperative period or blood transfusion during operation will have adverse effects (Liu et al., 2022). Similarly, the study of Van Cutsem et al. (2019) believes that active gastrointestinal bleeding is not a risk factor for gastrointestinal malignant tumors, and the increased severity of anemia will not pose a risk to the prognosis of patients with colorectal cancer. The conclusion of this study on the prognosis of preoperative anemia in patients with CRC is significantly different from that of the study of Van Cutsem et al. (2019). We found that the five-year survival rate of the non-anemia group was significantly higher than that of the anemia group, and the multivariate analysis showed that anemia was an independent risk factor affecting the prognosis of colorectal cancer patients. Most studies now believe that preoperative anemia is an independent risk factor for postoperative incidence rate and mortality in surgical patients, which is consistent with the results of this study (Böhm, Schmalzing & Meybohm, 2022). Anemia is a common and significant problem in patients with colorectal cancer (Ploug et al., 2022). Common causes include tumor hemorrhage, malnutrition, chronic inflammation, tumor infiltration of bone marrow, reduction of erythropoietin synthesis, bone marrow suppression caused by cancer treatment among others (Sawayama et al., 2021). The study also found that tumor size, TNM stage, distant metastasis, mGPS score, and Ki-67 were independent risk factors affecting the prognosis of colorectal cancer patients. Other studies have shown that anemia, even mild anemia, is a risk factor for postoperative complications and more extended hospital stays in colorectal cancer patients (Liu et al., 2018). Gaspar, Sharma & Das (2015) believe that anemia will lead to increased tumor hypoxia and increased HIF-1 α Expression and promote the transcription of target genes involved in angiogenesis, proliferation, and metastasis (Voit & Sankaran, 2020). Because intraoperative bleeding may aggravate anemia, correcting preoperative anemia is more critical for colorectal cancer patients undergoing surgery. However, this study also has some limitations. As this study is a single-center retrospective study with a small sample size, a large-scale multicenter study still needs to verify the conclusion.

Conclusions

Anemia is common in colorectal cancer patients. The oncological characteristics of colorectal cancer patients with different types of preoperative anemia are different. Preoperative anemia is an independent risk factor for the prognosis of patients with colorectal cancer. Therefore, clinicians should correct patients’ anemia and systemic inflammation as early as possible in the stage of comprehensive management before operation, to provide a better prognosis for patients with colorectal cancer.

Supplemental Information

Data S1 Raw Data

Click here for additional data file.

I would like to express my gratitude to all those helped me during the writing of this thesis. I acknowledge the help of my colleagues, Hongqing Ma and Guanglin Wang They have offered me suggestion in academic studies.

Additional Information and Declarations

Competing Interests

Author Contributions

Human Ethics

Data Availability

The authors declare there are no competing interests.

Chaoxi Zhou conceived and designed the experiments, analyzed the data, prepared figures and/or tables, and approved the final draft.

Hongqing Ma conceived and designed the experiments, analyzed the data, prepared figures and/or tables, and approved the final draft.

Guanglin Wang performed the experiments, authored or reviewed drafts of the article, and approved the final draft.

Youqiang Liu performed the experiments, authored or reviewed drafts of the article, and approved the final draft.

Baokun Li performed the experiments, authored or reviewed drafts of the article, and approved the final draft.

Jian Niu conceived and designed the experiments, authored or reviewed drafts of the article, and approved the final draft.

Yang Zhao performed the experiments, analyzed the data, prepared figures and/or tables, and approved the final draft.

Guiying Wang conceived and designed the experiments, authored or reviewed drafts of the article, and approved the final draft.

The following information was supplied relating to ethical approvals (i.e., approving body and any reference numbers):

All samples obtained in this study were approved by the ethics committee of the Fourth Hospital of Hebei Medical University and abided by the ethical guidelines of the Declaration of Helsinki.

The following information was supplied regarding data availability:

The raw data is available in the Supplementary File.

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
