# Peer review of "Association between different types of preoperative anemia and tumor characteristics, systemic inflammation, and survival in colorectal cancer"

_PeerJ, doi:10.7717/peerj.16293_

## Round 0.1 · original submission · Major Revisions

Please respond and make appropriate revisions based on the reviewers' suggestions and my comments (below). This will greatly improve the quality of the manuscript.

Here are my comments:
1. Firstly, it would be helpful to provide more specific information regarding the incidence of anemia in CRC patients, especially in relation to different stages of the disease. This would provide a clearer understanding of the extent of the problem.
2. Secondly, the introduction mentions the lack of research on the relationship between different types of anemia and prognosis in CRC patients before surgery. It would be helpful to explain the significance of studying this relationship and how it could potentially impact treatment decisions or patient outcomes.
3. Line 116: Add [and] before Ki-67.
4. Line 156: [were] should be [was].
5. Lines 208-210: The limitations of this study should be displayed in the Discussion section, but not in the Conclusion section.

**Language Note:** PeerJ staff have identified that the English language needs to be improved. When you prepare your next revision, please either (i) have a colleague who is proficient in English and familiar with the subject matter review your manuscript, or (ii) contact a professional editing service to review your manuscript. PeerJ can provide language editing services - you can contact us at copyediting@peerj.com for pricing (be sure to provide your manuscript number and title). – PeerJ Staff

Reviewer 1 ·

Basic reporting

Overall, the manuscript presents a valuable study exploring the association between different types of preoperative anemia and tumor characteristics, systemic inflammation, and survival in colorectal cancer.
The study design, data analysis, and presentation of results are generally well-executed. However, there are a few issues that need revision and improvement.
1. The abstract might indicate its implications for clinical practice.
2. The introduction might also offer a concise indication of the current evidence on this topic and recognize the gaps in knowledge that the study purposes to report.
3. [95 patients were included in the study, including 54 males and 41 females.] should be revised to [Ninety-five patients were included in the study, including 54 males and 41 females].
4. Describe the inclusion and exclusion criteria used for selecting eligible individuals in this study.

Experimental design

5. Clarify if the evaluation of preoperative anemia status was done on a study population or between different study groups.
6. Explain why the Hb, MCV, MCH, and MCHC were chosen to group patients and provide a brief justification for its use in this analysis.

Validity of the findings

7. Clearly state the primary observation and the secondary observation in the method section in this study.

Additional comments

8. Define the terms " mGPS scores " and " Ki-67 positive expression rate " to ensure readers have a clear understanding of these specialized terms.
9. The authors should provide a clear rational for each analysis as there is a lack of integration.

Reviewer 2 ·

Basic reporting

good

Experimental design

good

Validity of the findings

good

Additional comments

The summary of the author is not good, and it is suggested to be revised. Not only that, but there are also many problems, which need to be revised seriously by the author.
1) Provide a brief overview of the Cox proportional hazard regression and why it was deemed suitable for evaluating prognostic factors of patients.
2) The statement on preoperative anemia in CRC patients’ needs to be supported by specific references or evidence.
3) Provide a rationale, detailed operation procedure for using multivariate analysis for the study.
4) Consider reporting the PRISM software versions used for the analysis to ensure transparency and reproducibility.
5) I think authors did not correctly cite the references. For example, reference 21 was current insights regarding epidemiology, cancer, and DNA repair, which did not match the content of the article.
6) Provide a brief overview of the using the Dunnett's post‐hoc, and its purpose in assessing the statistical significance among the three groups. How was statistical significance determined for the analysis?

Reviewer 3 ·

Basic reporting

This is a very interesting article, and we have reviewed it to confirm that it is worthy of publication, but there are still some issues, please revise the issues below.
1. Full name for the abbreviation may be presented at the first presence.
2. Line 51, many studies have reported the association between anemia and poor prognosis in colorectal cancer patients. Is there any innovation in this article compared to the previous article?
3. The data shown here are rather underwhelming, to be honest, and it is not clear on which basis studies/reports were included and why others (a lot of studies, in fact) have been excluded. It is described in materials & methods, how studies were included/excluded, and it is difficult for a reviewer to appreciate this without performing a lengthy literature study by himself.

Experimental design

No

Validity of the findings

No

Additional comments

4. English language should be revised by a professional English polishing.
5. The bias and limitations need to be discussed in the discussion part.

---

## Round 0.2 · Minor Revisions

There are still several issues that need to be revised:

1. Table 1, Item: Table 1 was inappropriately compressed and the number of patients in each group cannot be seen by the readers. [anemia(n=52), anemia(n=14), and anemia (n=29)] should be clearly presented in Table 1.

2. Delete the contents shown in lines 155 and 156, because it seems to be an error that occurs during text editing.

3. Line 140 [tumor site] should be [tumor size]. Please correct it.

4. Line 147 [CA724 and mGPS scores among the three groups (P<0.05, Table 3)]: There was no significant difference in CA724 levels between those three groups (P= 0.291)? Please carefully confirm this.

5. Line 147 [CEA, CA199, CA242, and CA125 (P>0.05, Table 1)]: These results were shown in Table 3, but not in Table 1. Please carefully confirm this and make the necessary revisions.

6. Conclusion section: [and systemic inflammation may be related to the status of preoperative anemia] could be deleted here, because This expression of uncertainty affects the solidity of the conclusion.

Reviewer 1 ·

Basic reporting

Based on my review comments, the author has made good revisions to the abstract, introduction, experimental design, and results sections. I believe that the article has met the publication standards.

Experimental design

In the experimental design section, the author made corresponding modifications based on my review comments, and the modifications were very good.

Validity of the findings

In the methodology section of this study, the author provides a clear explanation of the main and secondary observations.

Additional comments

No comment.

Reviewer 2 ·

Basic reporting

The author have added the detailed statements on preoperative anemia in CRC patients in the introduction and discussion section, and multivariate analysis was carried out by the Cox's regression multiple hazard model.The author has made good revisions to my review comments.

Experimental design

The experimental design section of the article has been reasonably revised.

Validity of the findings

The results section of the article has been greatly improved.

Additional comments

The author has made detailed revisions and responses to my review comments, and I have no further comments. I believe that this article has met the standards for publication.

Reviewer 3 ·

Basic reporting

The article has been standardized and revised according to my review comments, and I have no further comments.

Experimental design

Well modified. I have no further comments.

Validity of the findings

Well modified. I have no further comments.

Additional comments

After the author revised my review comments one by one, this manuscript has been greatly strengthened. I have no further suggestions, the article is sufficient to meet the publication standards of the magazine.

---

## Round 0.3 · accepted · Accept

Most of my concerns were addressed.

However, it should be noted that in the Abstract on the journal's web page (Version 2) and in the Abstract of Manuscript_(clean).docx, the sentence (below) that I suggested to be changed was still not deleted, which is probably because the authors only revised the Conclusion and forgot to revise the Abstract.

[, and systemic inflammation may be related to the status of preoperative anemia]